# The Synergistic Inhibitions of Tungstate and Molybdate Anions on Pitting Corrosion Initiation for Q235 Carbon Steel in Chloride Solution

**DOI:** 10.3390/ma15248986

**Published:** 2022-12-16

**Authors:** Huanggen Yang, Pei Zhang, Guochao Nie, Yong Zhou

**Affiliations:** 1Key Laboratory of Coordination Chemistry of Jiangxi Province, College of Chemistry and Chemical Engineering, Jinggangshan University, Ji’an 343009, China; 2Guangxi Key Lab of Agricultural Resources Chemistry and Biotechnology, College of Chemistry and Food Science, Yulin Normal University, Yulin 537000, China; 3Key Laboratory of Green Chemical Process of Ministry of Education, School of Chemical Engineering and Pharmacy, Wuhan Institute of Technology, Wuhan 430205, China

**Keywords:** Q235 carbon steel, pitting corrosion, tungstate (WO_4_^2−^), molybdate (MoO_4_^2−^), passive film

## Abstract

In this work, the synergistic inhibitions of tungstate (WO_4_^2−^) and molybdate (MoO_4_^2−^) anions, including role and mechanism, on the initiation of pitting corrosion (PC) for Q235 carbon steel in chloride (Cl^−^) solution were investigated with electrochemical and surface techniques. The pitting potential (E_p_) of the Q235 carbon steel in WO_4_^2−^ + MoO_4_^2-^ + Cl^−^ solution was more positive than that in WO_4_^2−^ + Cl^−^ or MoO_4_^2−^ + Cl^−^ solution; at each E_p_, both peak potential and affected region of active pitting sites in WO_4_^2−^ + MoO_4_^2−^ + Cl^−^ solution were smaller than those in WO_4_^2−^ + Cl^−^ or MoO_4_^2−^ + Cl^−^ solution. WO_4_^2−^ and MoO_4_^2−^ showed a synergistic role to inhibit the PC initiation of the Q235 carbon steel in Cl^−^ solution, whose mechanism was mainly attributed to the influences of two anions on passive film. Besides iron oxides and iron hydroxides, the passive film of the Q235 carbon steel formed in WO_4_^2−^ + Cl^−^, MoO_4_^2−^ + Cl^−^, or WO_4_^2−^ + MoO_4_^2−^ + Cl^−^ solution was also composed of FeWO_4_ plus Fe_2_(WO_4_)_3_, Fe_2_(MoO_4_)_3_, or Fe_2_(WO_4_)_3_ plus Fe_2_(MoO_4_)_3_, respectively. The film resistance and the defect quantity for Fe_2_(WO_4_)_3_ plus Fe_2_(MoO_4_)_3_ film were larger and smaller than those for FeWO_4_ plus Fe_2_(WO_4_)_3_ film and Fe_2_(MoO_4_)_3_ film, respectively; for the inhibition of PC initiation, Fe_2_(WO_4_)_3_ plus Fe_2_(MoO_4_)_3_ film provided better corrosion resistance to Q235 carbon steel than FeWO_4_ plus Fe_2_(WO_4_)_3_ film and Fe_2_(MoO_4_)_3_ film did.

## 1. Introduction

Pitting corrosion (PC) is one of the most common and universal localized forms of corrosion for metals and alloys, and the remarkable characteristic of PC is latent, random and sudden, particularly during the process of PC initiation [1]. Therefore, it is generally accepted that the key point of PC inhibition is restraining the nucleation of the corrosion pit [2].

It is confirmed that for metals and alloys working in service environments, there are two essential factors for the initiation of PC: the establishment of surface passivation on metals and alloys and the presence of aggressive anions in service environments, particularly the chloride anion (Cl^−^) [1]. Further, it is reported repeatedly that in the service environments of metals and alloys, the addition of an introduced substance, mainly some inorganic and organic species, can inhibit the initiation of PC effectively [3]. At present, reported inorganic species for the inhibition of PC initiation include borate (BO_4_^3−^) [4], tetraborate (B_4_O_7_^2−^) [5], chromate (CrO_4_^2−^) [6], dichromate (Cr_2_O_7_^2−^) [7], molybdate (MoO_4_^2−^) [8], nitrite (NO_2_^−^) [9], nitrate (NO_3_^−^) [6], phosphate (PO_4_^3−^) [6], tripolyphosphate (P_3_O_10_^5−^) [10], tungstate (WO_4_^2−^) [11] anions and so on; by contrast, reported organic species for inhibiting PC initiation are relatively more, such as benzotriazole (BTA) [12], calcium lignosulfonate (CLS) [13], imidazoline (IM) [14], octaphenylpolyoxyethyiene (OP) [15], sodiumdodecylbenzenesulfonate (SDBS) [16], sodium dodecyl sulfate (SDS) [17], sodium oleate (SO) [18], thioureido imidazoline (TAI) [19], tetrabethylenepentamine (TEPA) [20], vitamin B5 [21] and others. However, in recent years, the research of PC inhibition has begun to focus on the synergistic role and mechanism of two or more species, and the conclusion that the simultaneous additions of some species into environmental media exhibit a better role for PC inhibition than the single addition of corresponding one species do has been proposed [22,23,24,25,26,27,28,29,30,31]. Up to now, confirmed combinations owning a synergistic role for PC inhibition are as follows: B_2_O_7_^2−^ and PO_4_^3−^ [22], Cr_2_O_7_^2−^ and MoO_4_^2−^ [23], MoO_4_^2−^ and NO_2_^−^ [24], MoO_4_^2−^ and BTA [25], MoO_4_^2−^ and CLS [26], NO_2_^−^ and TAI [27], NO_2_^−^ and TEPA [28], WO_4_^2−^ and CLS [29], CLS and SO [30], IM and OP [31], and so on.

Carbon steel is a kind of metal engineering material with relatively weak surface passivation capability; therefore, in pure chloride solution, general corrosion is the main type of corrosion damage for carbon steel [2]. Further, for carbon steel in pure chloride solution, it is reported largely that the additions of organic species and their adsorption role as well as the additions of inorganic species and their film-forming role can inhibit the general corrosion of carbon steel [32]. At the same time, it is also confirmed that if an inorganic species, whose inhibition against general corrosion is derived from its role for the surface passivation of carbon steel, its addition into chloride solution can induce the initiation of PC for carbon steel. The related report of the above result has been mentioned in Na_3_BO_4_ + NaCl solution [4], Na_3_PO_4_ + NaCl solution [6], NaNO_2_ + NaCl solution [9], Na_2_WO_4_ + NaCl solution [11], Na_2_MoO_4_ + NaCl solution [25] and so on. Further, from the above reports, it is concluded that for carbon steel in chloride solution containing inorganic species, particularly inorganic species with a certain degree of oxidation ability, the occurrence of PC for carbon steel results from the combined actions of Cl^−^ and inorganic species, and their appropriate proportion plays a critical role in the initiation of PC.

In our previous works [33,34,35], for Q235 carbon steel in pure Na_2_WO_4_ or/and Na_2_MoO_4_ solutions free of Cl^−^, we investigated its electrochemical behavior and surface passivation and summarized some rules about the influences of WO_4_^2−^ and MoO_4_^2−^ on its corrosion and passivation behaviors. Both WO_4_^2−^ and MoO_4_^2−^ could promote the surface passivation of the Q235 carbon steel, which was due to their influence on the composition and microstructure of the passive film. However, due to the absence of an aggressive anion, particularly Cl^−^, in pure Na_2_WO_4_ or/and Na_2_MoO_4_ solutions, the occurrence of PC for Q235 carbon steel is avoided [36]. Further, in the presence of Cl^−^, for carbon steel, in Na_2_WO_4_ + NaCl solution, Gao et al. [11] reported that the inhibition of WO_4_^2−^ on the initiation of PC was attributed to its role in promoting the formation of γ-Fe_2_O_3_, which was the main composition of the passive film on the carbon steel surface. Jabeera et al. [37] reported that the presence of WO_4_^2−^ resulted in the formation of FeWO_4_, which repaired the defect of the passive film because of the preferential deposition of FeWO_4_ at the defect sites. Fujioka et al. [38] reported that the pitting potential and repassivation potential of carbon steel moved to the positive direction with the increase inWO_4_^2−^ concentration, and the role of WO_4_^2−^ resulted from repairing the defect of the passive film and from inhibiting the development of the corrosion pit. At the same time, for carbon steel in Na_2_MoO_4_ + NaCl solution, Zhao et al. [23] reported that the presence of MoO_4_^2−^ restrained the nucleation and development of the corrosion pit, which was attributed to the pH value of the pit interior raised with the increase inMoO_4_^2−^ concentration. Zhou et al. [25] reported that the inhibition of MoO_4_^2−^ on PC initiation was due to the role of MoO_4_^2−^ to promote the transformation from FeOOH to Fe_2_O_3_ in the passive film and to further enhance the stability of the passive film. Fujioka et al. [38] also reported the influence of MoO_4_^2−^ concentration on the pitting potential and repassivation potential and its role in repairing the passive film defect. Saremi et al. [39] reported that the adsorption and reduction of MoO_4_^2−^ was beneficial to the high resistance and low permeability of the passive film.

As stated above, in the single presence of WO_4_^2−^ or MoO_4_^2−^ in chloride solution, its role and mechanism on the PC inhibition of carbon steel has been greatly reported; however, the synergistic role and mechanism of WO_4_^2−^ and MoO_4_^2−^ on PC inhibition, particularly on the inhibition of PC initiation, of carbon steel are absent. In this work, Na_2_WO_4_, Na_2_MoO_4_ and NaCl are introduced into de-ionized water to obtain three solutions: WO_4_^2−^ + Cl^−^ solution, MoO_4_^2−^ + Cl^−^ solution and WO_4_^2−^ + MoO_4_^2−^ + Cl^−^ solution. In the above three solutions, the synergistic inhibitions of WO_4_^2−^ and MoO_4_^2−^, including the inhibitive role and mechanism, on the initiation of PC for Q235 carbon steel are investigated with electrochemical and surface techniques.

## 2. Materials and Methods

The investigated material of this work was Q235 carbon steel with the following chemical composition (weight percent): C, 0.160; Mn, 0.530; P, 0.015; S, 0.045; Si, 0.300; and Fe, balance. Q235 carbon steel was processed into some samples with the three-dimensional size of 10 × 10 × 3 mm; after that, all samples were manually abraded up to 1000 grit with SiC abrasive paper, rinsed with de-ionized water and degreased in alcohol.

There were three solutions investigated in this work: besides Na^+^, Solution I comprising WO_4_^2−^ and Cl^−^, Solution II comprising MoO_4_^2−^ and Cl^−^, and Solution III comprising WO_4_^2−^, MoO_4_^2−^ and Cl^−^. The detailed information of the three solutions, including component and pH value, are listed in Table 1. Solutions I, II and III were prepared with analytical grade agents and de-ionized water.

The measurements of electrochemical techniques, including open circuit potential (OCP) evolution, potentiodynamic polarization, electrochemical impedance spectroscopy (EIS) and Mott–Schottky plot, were carried out by a Princeton 2273 electrochemical workstation (USA) at room temperature (RT). A typical three-electrode system was applied for electrochemical measurements: working electrode was a sample of the Q235 carbon steel, counter electrode was a platinum sheet, and reference electrode was a saturated calomel electrode (SCE). Before each electrochemical test, the surface area of the working electrode (a sample of the Q235 carbon steel) was restricted into a square with the two-dimensional size of 2 × 2 mm with room-temperature-cured silicone rubber. In OCP evolution tests, the record frequency of OCP was 5 Hz; in potentiodynamic polarization tests, the scanning rate of applied potential was 0.5 mV/s; in EIS tests, a perturbation potential of 10 mV amplitude was performed in the frequently range from 10^5^ to10^−2^ Hz; in Mott–Schottky plot tests, the scanning rate of applied potential was 5 mV/s, and the scanning range of applied potential was from −0.2 to 1.0 V_SCE_. However, all potentiodynamic polarization tests were terminated when corresponding current density increased suddenly and sharply, and thus, their potential scanning ranges were not proposed in this work.

The measurements of surface techniques included spatial potential distribution and surface chemical composition: a former test was performed by an XMU-BY electrochemical scanning tunneling microscope (ESTM) instrument (Xiamen Legang Materials Technology Co., Ltd., Xiamen, China), and a latter test was conducted by an ESCALAB-250 X-ray photoelectron spectroscopy (XPS) instrument (Waltham, MA, USA).

## 3. Results and Discussion

### 3.1. OCP Evolution

Figure 1 shows the OCP evolutions of the Q235 samples in WO_4_^2−^ + Cl^−^, MoO_4_^2−^ + Cl^−^ and WO_4_^2−^ + MoO_4_^2−^ + Cl^−^ solutions. For Q235 samples in three solutions, with the extension of test time, OCP moves to the positive direction from 0 to 10 min; after that, from 10 to 20 min, the change of OCP is slight. Therefore, prior to the subsequent electrochemical measurements of potentiodynamic polarization, EIS and Mott–Schottky plot, Q235 samples were immersed in corresponding solutions for 20 min to ensure the stability of OCP.

However, the stable OCP value of the Q235 carbon steel in WO_4_^2−^ + Cl^−^, MoO_4_^2−^ + Cl^−^ or WO_4_^2−^ + MoO_4_^2−^ + Cl^−^ solution increases in turn. For carbon steels in alkaline environments at open circuit condition, anodic half-reaction is the oxidation of an iron element from Fe to Fe^2+^ with the standard potential (*E_s_*) of −0.684 V_SCE_ [40]:Fe → Fe^2+^ + 2e(1)

The equilibrium potential (*E_e_*) of Fe oxidation is described as follows:*E_e_* (Fe^2+^/Fe) = (−0.684 + 0.059/2 log α_Fe2+_) V_SCE_(2)

At the same time, cathodic half-reaction is the reduction of an oxygen element from O_2_ to OH^−^ with the *E_s_* of −0.157 V_SCE_ [41]:O_2_ + 2H_2_O + 4e → 4OH^−^(3)

The *E_e_* of O_2_ reduction is described as follows:*E_e_* (O_2_/OH^−^) = (0.984 − 0.059 pH) V_SCE_(4)

In three solutions, their same pH value suggests the approximate *E_e_* of O_2_ reduction; therefore, according to Nernst theory, the difference in a stable OCP value is mainly attributed to the different *E_e_* of Fe oxidation. In the subsequent result of potentiodynamic polarization, it will be seen that in WO_4_^2−^ + Cl^−^, MoO_4_^2−^ + Cl^−^ or WO_4_^2−^ + MoO_4_^2−^ + Cl^−^ solution, the corrosion current density of the Q235 carbon steel decreases in turn, indicating that Fe^2+^ concentration near the solution/electrode interface decreases in turn. According to Equation (2), it can be calculated that in WO_4_^2−^ + Cl^−^, MoO_4_^2−^ + Cl^−^ or WO_4_^2−^ + MoO_4_^2−^ + Cl^−^ solution, the *E_e_* of Fe oxidation moves to the positive direction in turn, resulting in the stable OCP value of the Q235 carbon steel that increases in turn.

### 3.2. Potentiodynamic Polarization

Figure 2 shows the polarization curve of the Q235 sample in 0.1 mM NaCl solution and the cyclic polarization curves of the Q235 samples in WO_4_^2−^ + Cl^−^, MoO_4_^2−^ + Cl^−^ and WO_4_^2−^ + MoO_4_^2−^ + Cl^−^ solutions. For the Q235 sample in 0.1 mM NaCl solution, anodic current density increases persistently with the positive shift of applied potential, indicating that Q235 carbon steel presents the electrochemical behavior of activation in 0.1 mM NaCl solution [42]. In contrast, for Q235 samples in WO_4_^2−^ + Cl^−^, MoO_4_^2−^ + Cl^−^ and WO_4_^2−^ + MoO_4_^2−^ + Cl^−^ solutions, with the positive shift of applied potential, anodic current density firstly increases slowly from the corrosion potential (*E_c_*) to approximately −0.3 V_SCE_, then maintains steadily from −0.3 V_SCE_ to the pitting potential (*E_p_*), and finally increases rapidly when the applied potential is up to *E_p_*. Q235 carbon steel presents the electrochemical behavior of activation–passivation–pitting in WO_4_^2−^ + Cl^−^, MoO_4_^2−^ + Cl^−^ and WO_4_^2−^ + MoO_4_^2−^ + Cl^−^ solutions [43].

However, the influence of solution component on *E_c_* and corrosion current density (*i_c_*) is negligible, but on *E_p_* and repassivation potential (*E_r_*), it is very significant. Table 2 lists the calculated values of *E_c_*, *i_c_*, *E_p_* and *E_r_*, in which each *E_c_*, *i_c_*, *E_p_* or *E_r_* datum is the average value from ten parallel potentiodynamic polarization tests. For Q235 carbon steel in three solutions, the values of *E_p_* and (*E_p_* − *E_r_*) in WO_4_^2−^ + MoO_4_^2−^ + Cl^−^ solution are respectively larger and smaller than those in WO_4_^2−^ + Cl^−^ or MoO_4_^2−^ + Cl^−^ solution, indicating that WO_4_^2−^ and MoO_4_^2−^ show a synergistic role for the inhibition of PC initiation [44]. Because the main aim of this work is to reveal the synergistic role and mechanism of WO_4_^2−^ and MoO_4_^2−^ on the inhibition of PC initiation, the following discussion of this work is mainly focused on physical and chemical properties about the surface of the Q235 carbon steel at the applied potential of *E_p_*.

### 3.3. Spatial Potential Distribution (SPD)

In order to obtain spatial electrochemical information of PC initiation and propagation, the applied potentials of 0.01, 0.09 and 0.17 V_SCE_, respectively, being very close to the *E_p_* of the Q235 carbon steel in WO_4_^2−^ + Cl^−^, MoO_4_^2−^ + Cl^−^ and WO_4_^2−^ + MoO_4_^2−^ + Cl^−^ solutions, are exerted on the working electrode artificially.

Figure 3 shows the SPD images of the Q235 samples in WO_4_^2−^ + Cl^−^, MoO_4_^2−^ + Cl^−^ and WO_4_^2−^ + MoO_4_^2−^ + Cl^−^ solutions at the applied potential of 0.01 V_SCE_. At 0.01 V_SCE_, in the WO_4_^2−^ + Cl^−^ solution shown in Figure 3a, an active pitting site with the peak potential of 1.22 mV is detected at the bottom right corner of ESTM tip scanning region, suggesting the initiation of a corrosion pit [45]; by contrast, in the MoO_4_^2−^ + Cl^−^ solution shown in Figure 3b and in the WO_4_^2−^ + MoO_4_^2−^ + Cl^−^ solution shown in Figure 3c, no obvious active pitting sites are observed in the ESTM tip scanning region.

Figure 4 shows the SPD images of the Q235 samples in WO_4_^2−^ + Cl^−^, MoO_4_^2−^ + Cl^−^ and WO_4_^2−^ + MoO_4_^2−^ + Cl^−^ solutions at the applied potential of 0.09 V_SCE_. In WO_4_^2−^ + Cl^−^ solution, both the peak potential and affected region of the active pitting sites at 0.09 V_SCE_ are larger than those at 0.01 V_SCE_, as shown in Figure 4a and Figure 3a, indicating the end of PC initiation and the beginning of PC propagation [46]. At 0.09 V_SCE_, in the MoO_4_^2−^ + Cl^−^ solution shown in Figure 4b, an active pitting site with the peak potential of 1.16 mV is detected at the top right corner of the ESTM tip scanning region; in the WO_4_^2−^ + MoO_4_^2−^ + Cl^−^ solution shown in Figure 4c, no obvious active pitting sites are observed in the ESTM tip scanning region.

Figure 5 shows the SPD images of the Q235 samples in WO_4_^2−^ + Cl^−^, MoO_4_^2−^ + Cl^−^ and WO_4_^2−^ + MoO_4_^2−^ + Cl^−^ solutions at the applied potential of 0.17 V_SCE_. In WO_4_^2−^ + Cl^−^ solution, both the peak potential and affected region of the active pitting sites at 0.17 V_SCE_ further increase compared with those at 0.09 and 0.01 V_SCE_, as shown in Figure 3a, Figure 4a and Figure 5a, suggesting the continuous propagation of PC [47]. In MoO_4_^2−^ + Cl^−^ solution, both the peak potential and affected region of the active pitting sites at 0.17 V_SCE_ are larger than those at 0.09 V_SCE_, as shown in Figure 4b and Figure 5b; at 0.17 V_SCE_, in the WO_4_^2−^ + MoO_4_^2−^ + Cl^−^ solution shown Figure 5c, an active pitting site with the peak potential of 1.06 mV is detected at the middle position of the ESTM tip scanning region.

From the above results of polarization and SPD, on the one hand, the *E_p_* of the Q235 carbon steel in WO_4_^2−^ + MoO_4_^2−^ + Cl^−^ solution is more positive than that in WO_4_^2−^ + Cl^−^ or MoO_4_^2−^ + Cl^−^ solution; on the other hand, at each *E_p_*, both the peak potential and affected region of the active pitting sites in WO_4_^2−^ + MoO_4_^2−^ + Cl^−^ solution are smaller than those in WO_4_^2−^ + Cl^−^ or MoO_4_^2−^ + Cl^−^ solution. The above two aspects conclude that for Q235 carbon steel in Cl^−^ solution, the inhibitive role of WO_4_^2−^ and MoO_4_^2−^ on PC initiation when both are used together is better than that when one is used solely.

### 3.4. XPS

Figure 6 shows the wide-scan XPS of the Q235 samples polarized to *E_p_* in WO_4_^2−^ + Cl^−^, MoO_4_^2−^ + Cl^−^ and WO_4_^2−^ + MoO_4_^2−^ + Cl^−^ solutions. For Q235 samples in three solutions, four peaks, Fe 2p at about 711 eV, O 1s at about 531 eV, C 1s at about 286 eV and Fe 3p at about 55 eV, are gathered by XPS analysis, which is independent of the solution component. Besides the above XPS peaks, W 4d peak at about 247 eV as well as W 4f peak at about 36 eV in WO_4_^2−^ + Cl^−^ solution, Mo 3d peak at about 235 eV in MoO_4_^2−^ + Cl^−^ solution, and W 4d peak at about 247 eV, Mo 3d peak at about 235 eV as well as W 4f peak at about 37 eV in WO_4_^2−^ + MoO_4_^2−^ + Cl^−^ solution are also gathered by XPS analysis. The result of wide-scan XPS indicates that the passive film of the Q235 carbon steel formed in the three solutions has different chemical composition.

Figure 7 shows the high-resolution XPS of Fe 2p for Q235 samples polarized to *E_p_* in WO_4_^2−^ + Cl^−^, MoO_4_^2−^ + Cl^−^ and WO_4_^2−^ + MoO_4_^2−^ + Cl^−^ solutions. In the WO_4_^2−^ + Cl^−^ solution shown in Figure 7a, the Fe 2p spectrum of the Q235 sample reveals three peaks at 712.05, 710.85 and 706.95 eV, respectively, corresponding to the Fe element in the valence states of Fe^3+^, Fe^2+^ and Fe^0^ [48]; in MoO_4_^2−^ + Cl^−^ solution shown in Figure 7b and in WO_4_^2−^ + MoO_4_^2−^ + Cl^−^ solution shown in Figure 7c, each Fe 2p spectrum also reveals three peaks corresponding to Fe^3+^, Fe^2+^ and Fe^0^. However, the intensity of the Fe^3+^ peak in MoO_4_^2−^ + Cl^−^ and WO_4_^2−^ + MoO_4_^2−^ + Cl^−^ solutions is obviously stronger than that in WO_4_^2−^ + Cl^−^ solution; conversely, the intensities of Fe^2+^ and Fe^0^ are weaker in MoO_4_^2−^ + Cl^−^ and WO_4_^2−^ + MoO_4_^2−^ + Cl^−^ solutions than in WO_4_^2−^ + Cl^−^ solution. This result implies that the oxidation ability of MoO_4_^2−^ is stronger than that of WO_4_^2−^, which is also reported regarding their oxidation for stainless steel [49] and cold rolling steel [50], similarly.

In this work, the relatively strong oxidation ability of MoO_4_^2−^ plays an important role in the synergistic inhibitions of WO_4_^2−^ and MoO_4_^2−^. In the subsequent results of EIS and the Mott–Schottky plot, it will be seen that the passive film of the Q235 carbon steel formed in WO_4_^2−^ + MoO_4_^2−^ + Cl^−^ solution has larger passive film resistance and a smaller passive film defect than that formed in WO_4_^2−^ + Cl^−^ or MoO_4_^2−^ + Cl^−^ solution, which is very closely associated with the relatively high content of Fe^3+^ in the passive film. On the other hand, according to bipolar model [51], more Fe^3+^ and less Fe^2+^ transfer across the passive film from an anion-selective type to a cation-selective type, which is beneficial to prevent passive film from the adsorption and attack of Cl^−^.

Figure 8 shows the high-resolution XPS of W 4f for Q235 samples polarized to *E_p_* in WO_4_^2−^ + Cl^−^ and WO_4_^2−^ + MoO_4_^2−^ + Cl^−^ solutions. The W 4f spectrum of the Q235 sample exhibits the spin-orbit splitting double peaks of 37.50 eV as well as 35.40 eV in WO_4_^2−^ + Cl^−^ solution and 37.55 eV as well as 35.15 eV in WO_4_^2−^ + MoO_4_^2−^ + Cl^−^ solution, indicating that the W element is present in the passive film of the Q235 carbon steel formed in WO_4_^2−^ + Cl^−^ and WO_4_^2−^ + MoO_4_^2−^ + Cl^−^ solutions.

Figure 9 shows the high-resolution XPS of Mo 3d for Q235 samples polarized to *E_p_* in MoO_4_^2−^ + Cl^−^ and WO_4_^2−^ + MoO_4_^2−^ + Cl^−^ solutions. Similar to W 4f, Mo 3d also exhibits spin-orbit splitting double peaks: at 235.70 and 232.40 eV in MoO_4_^2−^ + Cl^−^ solution, and at 235.80 and 232.70 eV in WO_4_^2−^ + MoO_4_^2−^ + Cl^−^ solution.

When carbon steels are exposed in an atmosphere environment, an air-formed passive film forms on their surface spontaneously [52]; further, when served in alkaline environments, previous passive film formed in air can rearrange a double-layer microstructure [53]. The inner dense layer of the passive film is composed of FeOOH and Fe_2_O_3_, and the outer loose layer of the passive film is composed of Fe(OH)_2_·nH_2_O and Fe(OH)_3_·nH_2_O [54]. The related mechanism of the above statement is as follows [55]:Fe + OH^−^ → FeOH^−^_ads_(5)
FeOH^−^_ads_ → FeOH_ads_ + e(6)
FeOH_ads_ + OH^−^ → Fe(OH)_2_ + e(7)
Fe(OH)_2_ + OH^−^ → FeOOH + H_2_O + e(8)

Although there are many defects in the outer loose layer of the passive film [56], some introduced substances, such as WO_4_^2−^ and MoO_4_^2−^, can repair them [49,50]. According to the present XPS results and our previous works [33,34,35], it is concluded that besides iron oxides and iron hydroxides, the passive film of the Q235 carbon steel formed in WO_4_^2−^ + Cl^−^, MoO_4_^2−^ + Cl^−^, or WO_4_^2−^ + MoO_4_^2−^ + Cl^−^ solution also comprises FeWO_4_ plus Fe_2_(WO_4_)_3_, Fe_2_(MoO_4_)_3_, or Fe_2_(WO_4_)_3_ plus Fe_2_(MoO_4_)_3_, respectively. The detailed composition of the passive film on the surface of the Q235 carbon steel formed in three solutions is listed in Table 3.

### 3.5. EIS

Figure 10 shows the EIS of the Q235 samples in 0.1 mM NaCl solution and in WO_4_^2−^ + Cl^−^, MoO_4_^2−^ + Cl^−^ and WO_4_^2−^ + MoO_4_^2−^ + Cl^−^ solutions. For the Q235 sample in 0.1 mM NaCl solution, the corresponding EIS is composed of only one capacitive semicircle in the entire frequency zone; in contrast, for Q235 samples in WO_4_^2−^ + Cl^−^, MoO_4_^2−^ + Cl^−^ and WO_4_^2−^ + MoO_4_^2−^ + Cl^−^ solutions, each EIS is composed of a relatively small capacitive semicircle in the high-frequency zone and another relatively large capacitive semicircle in the low-frequency zone. In the passivation system of the electrode/electrolyte, the appearance of the capacitive semicircle in EIS mainly results from the formation of the passive film on the interface between the electrode and electrolyte [57]. The present EIS results of the two capacitive semicircles confirm the double-layer microstructure of the passive film, and it is reasonable to infer that the small capacitive semicircle of EIS is attributed to the inner dense layer of the passive film, and another large one is due to the outer loose layer [35].

However, the influence of the solution component on the radius of the capacitive semicircle in the high-frequency zone is slight, but on that in the low-frequency zone, it is severe. That is to say, for the Q235 carbon steel in three solutions, the inhibitive role and mechanism of WO_4_^2−^ and MoO_4_^2−^ on PC initiation are mainly derived from their influence on the outer loose layer of the passive film, rather than to the inner dense layer. The radius of the large capacitive semicircle in WO_4_^2−^ + Cl^−^, MoO_4_^2−^ + Cl^−^ or WO_4_^2−^ + MoO_4_^2−^ + Cl^−^ solutions enlarges in turn, indicating that the corrosion resistance of the outer loose layer for the Fe_2_(WO_4_)_3_ plus Fe_2_(MoO_4_)_3_ film is better than that for the FeWO_4_ plus Fe_2_(WO_4_)_3_ film or Fe_2_(MoO_4_)_3_ film.

Further, the method of equivalent electrical circuit (EEC) fitting is applied to interpret EIS. Combining the EIS results shown in Figure 10 and our predecessors’ research studies [58], the model of EEC shown in Figure 11 is feasible to EIS interpretation. In Figure 11, *R_s_* represents solution resistance; *CPE_o_* and *R_o_* respectively represent the capacitance and resistance of the outer loose layer in the passive film; *CPE_i_* and *R_i_* represent the capacitance and resistance of the inner dense layer in the passive film, respectively.

Table 4 lists the fitted values of *CPE_o_*, *R_o_*, *CPE_i_* and *R_i_*, in which each *CPE_o_*, *R_o_*, *CPE_i_* or *R_i_* datum is the average value from ten parallel EIS tests. It was reported that the value of the passive film resistance reflected the anti-corrosion protection of the passive film, and the larger the resistance, the better the anti-corrosion protection [32]; the value of the passive film capacitance indicated the damaged area of the passive film, and the larger the capacitance, the more severe the film damage [57]. For Q235 carbon steel in WO_4_^2−^ + Cl^−^, MoO_4_^2−^ + Cl^−^ or WO_4_^2−^ + MoO_4_^2−^ + Cl^−^ solution, the values of *CPE_o_* and *R_o_* change obviously, but the values of *CPE_i_* and *R_i_* change slightly, confirming that for the passive film, the influence of WO_4_^2−^ and MoO_4_^2−^ on its outer loose layer is greater than that on its inner dense layer; on the other hand, the value of *CPE_o_* decreases, and the value of *R_o_* increases in turn, indicating that the Fe_2_(WO_4_)_3_ plus Fe_2_(MoO_4_)_3_ film shows better corrosion resistance against PC than the FeWO_4_ plus Fe_2_(WO_4_)_3_ film or Fe_2_(MoO_4_)_3_ film.

### 3.6. Mott–Schottky Plot

Figure 12 shows the Mott–Schottky plots of the Q235 samples in WO_4_^2−^ + Cl^−^, MoO_4_^2−^ + Cl^−^ and WO_4_^2−^ + MoO_4_^2−^ + Cl^−^ solutions. For the Q235 samples in three solutions, the slope of the straight line part for each Mott–Schottky plot exhibits a positive value, suggesting that the FeWO_4_ plus Fe_2_(WO_4_)_3_ film, Fe_2_(MoO_4_)_3_ film and Fe_2_(WO_4_)_3_ plus Fe_2_(MoO_4_)_3_ film satisfy the property of the n-type semiconductor [59]. It is generally accepted that for the passive film of n-type characteristic, its Mott–Schottky plot can be interpreted with the following equation [60]:*C*^−2^ = 2(*E* − *U_f_* − kT/e)/εε_0_e*N_D_*(9)

In Equation (9), *C* represents space charge layer capacitance, *E* represents applied potential, *U_f_* represents flat band potential, *k* is Boltzmann constant, *T* is absolute temperature, *e* is electron charge, *ε* represents passive film permittivity, *ε_0_* is free space permittivity, and *N_D_* represents donor density. Therein, the value of *N_D_* can reflect the defect quantity of the passive film: the larger the *N_D_* of the numerical value, the more defects in the passive film; at the same time, the value of *U_f_* can reflect the corrosion susceptibility of the working electrode in the electrolyte, similar to that of *E_c_* [61].

The influence of the solution component on *N_D_* is greater than that on *U_f_*, and Table 5 lists the fitted values of *N_D_* and *U_f_*, in which each *N_D_* or *U_f_* datum is the average valuefrom ten parallel Mott–Schottky plot tests. For Q235 carbon steel in WO_4_^2−^ + Cl^−^, MoO_4_^2−^ + Cl^−^ or WO_4_^2−^ + MoO_4_^2−^ + Cl^−^ solution, the value of *N_D_* increases in turn, indicating that the Fe_2_(WO_4_)_3_ plus Fe_2_(MoO_4_)_3_ film takes along smaller defects than the FeWO_4_ plus Fe_2_(WO_4_)_3_ film and Fe_2_(MoO_4_)_3_ film [61]; the value of *U_f_* changes slightly, which is similar to the value of *E_c_*, implying that Q235 carbon steel presents the approximate characteristic of uniform corrosion in three solutions.

In our predecessors’ research studies on PC, there are many theories and models proposed to reveal the initiation of a corrosion pit, such as acidification theory [62], chemical dissolution theory [63], depassivation–repassivation theory [64], anion penetration/migration model [65], chemical–mechanical model [66] and point defect model [67]. However, in the above theories and models [62,63,64,65,66,67], a critical step of PC initiation, the adsorption and the attack of aggressive anions on the surface of the passive film, was approved consistently by our predecessors. On the passive film, defective sites were more likely to be adsorbed and attacked by aggressive anions than other regions [1]. From the above results of EIS and Mott–Schottky plot, for the FeWO_4_ plus Fe_2_(WO_4_)_3_ film, Fe_2_(MoO_4_)_3_ film and Fe_2_(WO_4_)_3_ plus Fe_2_(MoO_4_)_3_ film, the film resistance (*R_i_*+*R_o_*) increases and the defect quantity decreases in turn. Therefore, the Fe_2_(WO_4_)_3_ plus Fe_2_(MoO_4_)_3_ film has a better capability to resist the adsorption and attack of Cl^−^ than FeWO_4_ plus Fe_2_(WO_4_)_3_ film or Fe_2_(MoO_4_)_3_ film [68], and WO_4_^2−^ and MoO_4_^2−^ exhibit a better inhibitive role on PC initiation when both are used together than when one is used solely.

## 4. Conclusions

(1)In Cl^−^ solution, the simultaneous additions of WO_4_^2−^ and MoO_4_^2−^ showed better inhibition on PC initiation than the single addition of WO_4_^2−^ or MoO_4_^2−^, and the synergistic role between WO_4_^2−^ and MoO_4_^2−^ for the inhibition of PC initiation was present.(2)Besides iron oxides and iron hydroxides, the passive film of Q235 carbon steel was also composed of FeWO_4_ plus Fe_2_(WO_4_)_3_ in WO_4_^2−^ + Cl^−^ solution, of Fe_2_(MoO_4_)_3_ in MoO_4_^2−^ + Cl^−^ solution and of Fe_2_(WO_4_)_3_ plus Fe_2_(MoO_4_)_3_ in WO_4_^2−^ + MoO_4_^2−^ + Cl^−^ solution.(3)The synergistic mechanism between WO_4_^2−^ and MoO_4_^2−^ for the inhibition of PC initiation was that the Fe_2_(WO_4_)_3_ plus Fe_2_(MoO_4_)_3_ film showed larger film resistance and smaller defect quantity than the FeWO_4_ plus Fe_2_(WO_4_)_3_ film and Fe_2_(MoO_4_)_3_ film.

## Figures and Tables

**Figure 1 materials-15-08986-f001:**
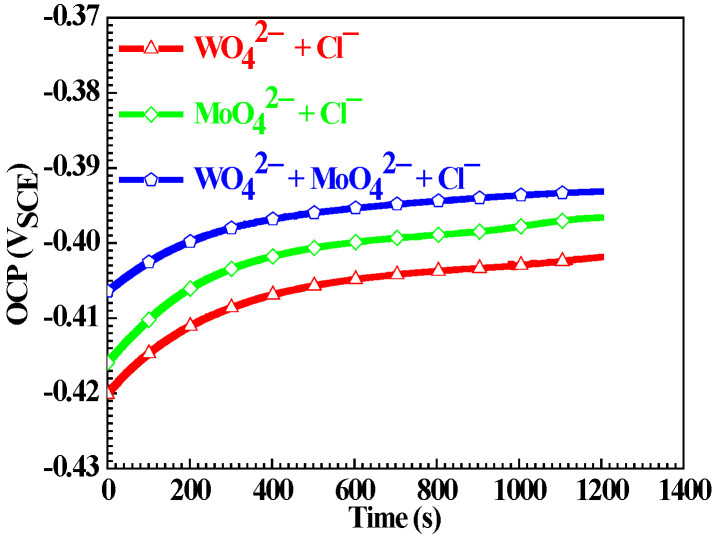
OCP evolutions of the Q235 samples in WO_4_^2−^ + Cl^−^, MoO_4_^2−^ + Cl^−^ and WO_4_^2−^ + MoO_4_^2−^ + Cl^−^ solutions.

**Figure 2 materials-15-08986-f002:**
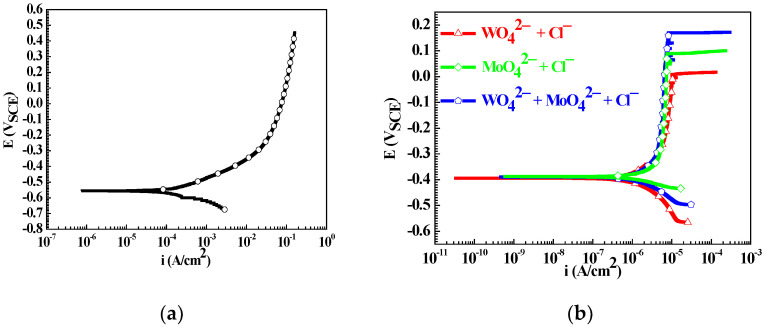
Polarization curve of the Q235 sample in 0.1 mM NaCl solution (**a**) and cyclic polarization curves of the Q235 samples in WO_4_^2−^ + Cl^−^, MoO_4_^2−^ + Cl^−^ and WO_4_^2−^ + MoO_4_^2−^ + Cl^−^ solutions (**b**).

**Figure 3 materials-15-08986-f003:**
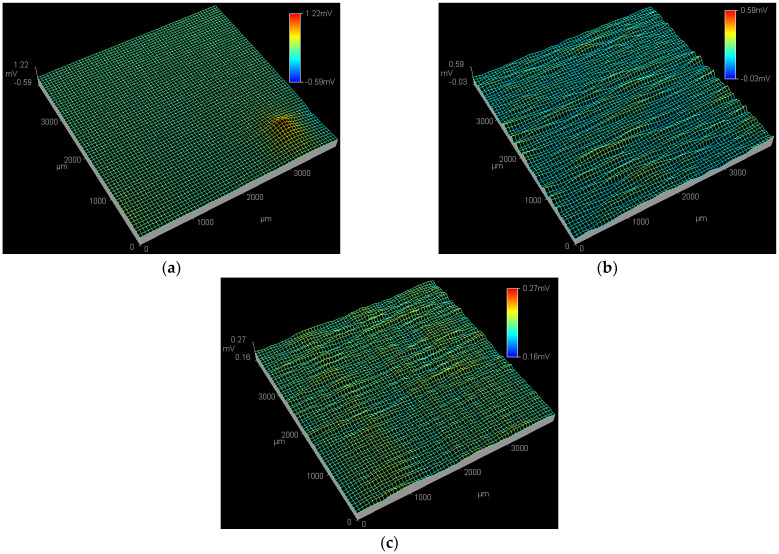
SPD images of the Q235 samples in WO_4_^2−^ + Cl^−^, MoO_4_^2−^ + Cl^−^ and WO_4_^2−^ + MoO_4_^2−^ + Cl^−^ solutions at applied potential of 0.01 V_SCE_: (**a**) WO_4_^2−^ + Cl^−^ solution, (**b**) MoO_4_^2−^ + Cl^−^ solution and (**c**) WO_4_^2−^ + MoO_4_^2−^ + Cl^−^ solution.

**Figure 4 materials-15-08986-f004:**
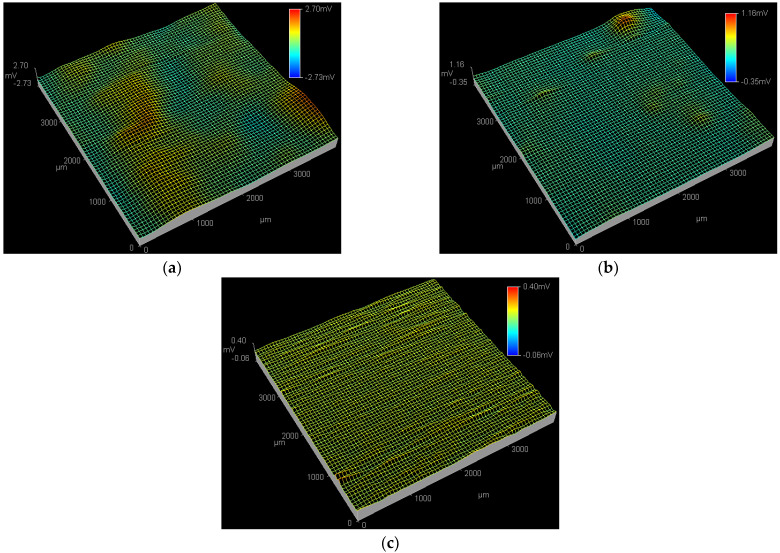
SPD images of the Q235 samples in WO_4_^2−^ + Cl^−^, MoO_4_^2−^ + Cl^−^ and WO_4_^2−^ + MoO_4_^2−^ + Cl^−^ solutions at applied potential of 0.09 V_SCE_: (**a**) WO_4_^2−^ + Cl^−^ solution, (**b**) MoO_4_^2−^ + Cl^−^ solution and (**c**) WO_4_^2−^ + MoO_4_^2−^ + Cl^−^ solution.

**Figure 5 materials-15-08986-f005:**
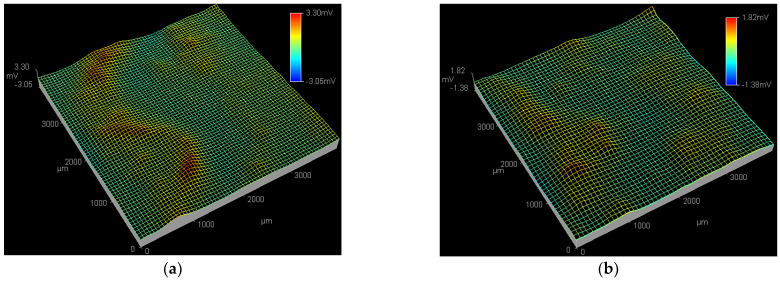
SPD images of the Q235 samples in WO_4_^2−^ + Cl^−^, MoO_4_^2−^ + Cl^−^ and WO_4_^2−^ + MoO_4_^2−^ + Cl^−^ solutions at applied potential of 0.17 V_SCE_: (**a**) WO_4_^2−^ + Cl^−^ solution, (**b**) MoO_4_^2−^ + Cl^−^ solution and (**c**) WO_4_^2−^ + MoO_4_^2−^ + Cl^−^ solution.

**Figure 6 materials-15-08986-f006:**
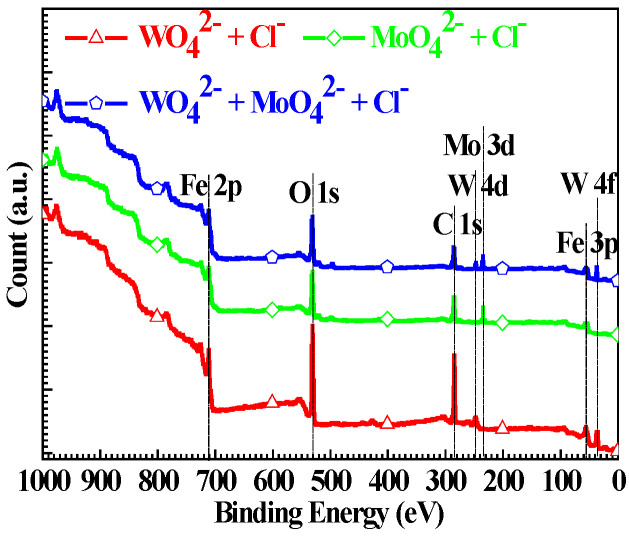
Wide-scan XPS of the Q235 samples polarized to *E_p_* in WO_4_^2−^ + Cl^−^, MoO_4_^2−^ + Cl^−^ and WO_4_^2−^ + MoO_4_^2−^ + Cl^−^ solutions.

**Figure 7 materials-15-08986-f007:**
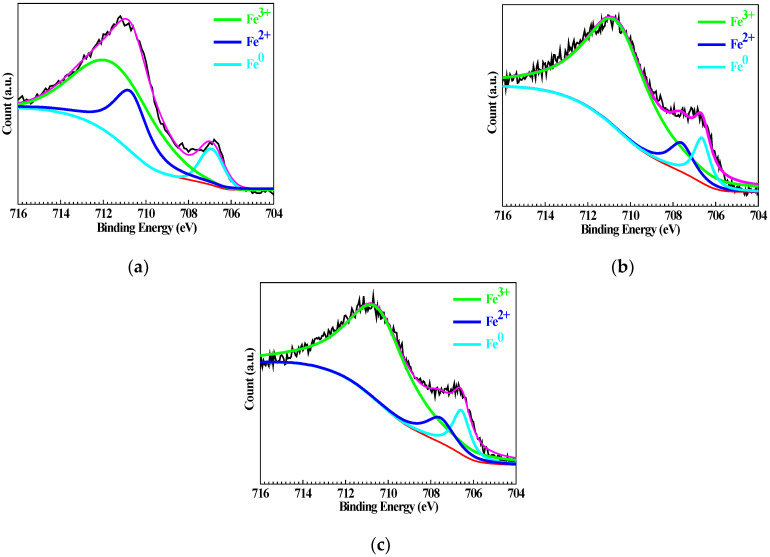
High-resolution XPS of Fe 2p for Q235 samples polarized to *E_p_* in WO_4_^2−^ + Cl^−^, MoO_4_^2−^ + Cl^−^ and WO_4_^2−^ + MoO_4_^2−^ + Cl^−^ solutions: (**a**) WO_4_^2−^ + Cl^−^ solution, (**b**) MoO_4_^2−^ + Cl^−^ solution and (**c**) WO_4_^2−^ + MoO_4_^2−^ + Cl^−^ solution.

**Figure 8 materials-15-08986-f008:**
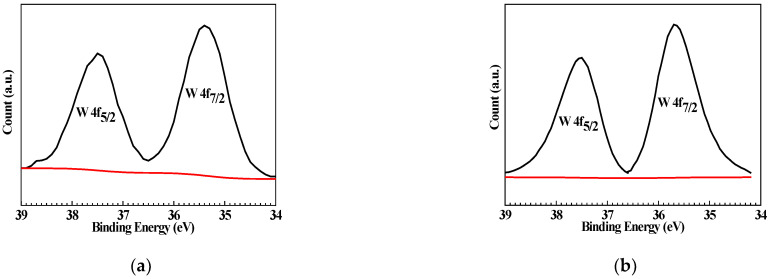
High-resolution XPS of W 4f for Q235 samples polarized to *E_p_* in WO_4_^2−^ + Cl^−^ and WO_4_^2−^ + MoO_4_^2−^ + Cl^−^ solutions: (**a**) WO_4_^2−^ + Cl^−^ solution and (**b**) WO_4_^2−^ + MoO_4_^2−^ + Cl^−^ solution.

**Figure 9 materials-15-08986-f009:**
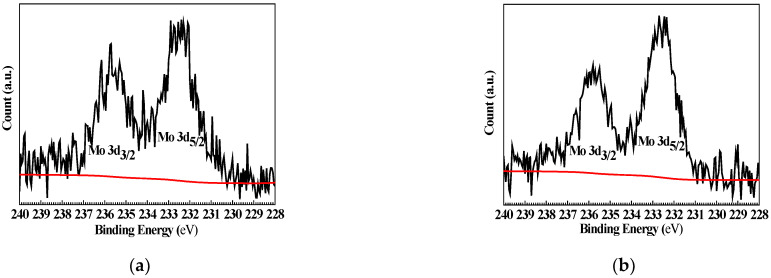
High-resolution XPS of Mo 3d for Q235 samples polarized to *E_p_* in MoO_4_^2−^ + Cl^−^ and WO_4_^2−^ + MoO_4_^2−^ + Cl^−^ solutions: (**a**) MoO_4_^2−^ + Cl^−^ solution and (**b**) WO_4_^2−^ + MoO_4_^2−^ + Cl^−^ solution.

**Figure 10 materials-15-08986-f010:**
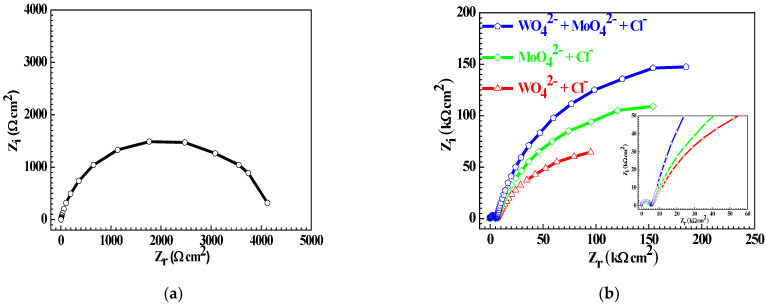
EIS of the Q235 samples in 0.1 mM NaCl solution (**a**) and in WO_4_^2−^ + Cl^−^, MoO_4_^2−^ + Cl^−^ and WO_4_^2−^ + MoO_4_^2−^ + Cl^−^ solutions (**b**).

**Figure 11 materials-15-08986-f011:**
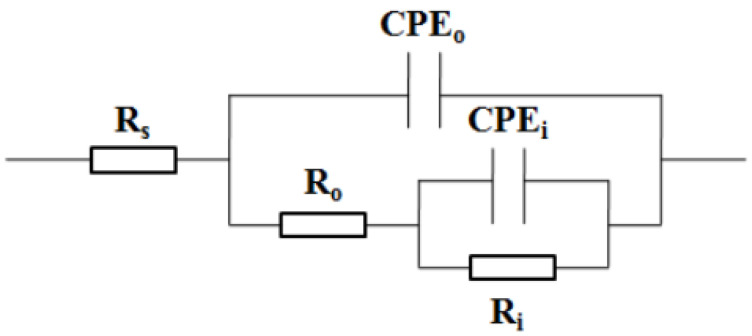
EEC model for EIS interpretation.

**Figure 12 materials-15-08986-f012:**
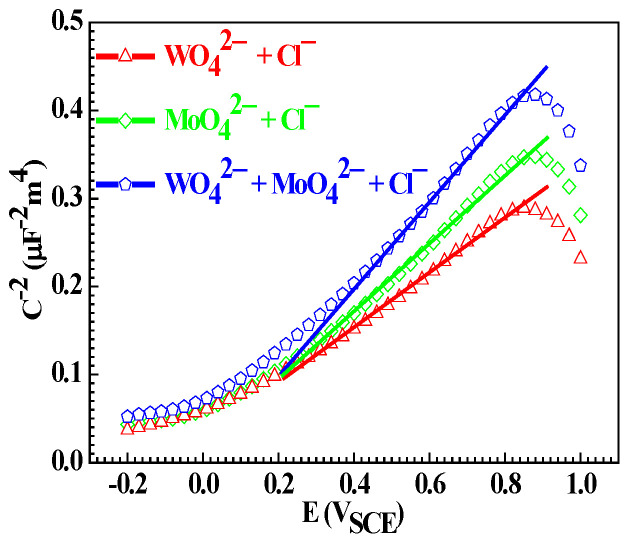
Mott–Schottky of the Q235 samples in WO_4_^2−^ + Cl^−^, MoO_4_^2−^ + Cl^−^ and WO_4_^2−^ + MoO_4_^2−^ + Cl^−^ solutions.

**Table 1 materials-15-08986-t001:** Component and pH value of Solution I, Solution II and Solution III.

Solution	Component	pH	Description
I	0.2 mM Na_2_WO_4_ + 0.1 mM NaCl	8.5	WO_4_^2−^ + Cl^−^
II	0.2 mM Na_2_MoO_4_ + 0.1 mM NaCl	8.5	MoO_4_^2−^ + Cl^−^
III	0.1 mM Na_2_WO_4_ + 0.1 mM Na_2_MoO_4_ + 0.1 mM NaCl	8.5	WO_4_^2−^ + MoO_4_^2−^ + Cl^−^

**Table 2 materials-15-08986-t002:** Calculated values of *E_c_*, *i_c_*, *E_p_*, *E_r_* and (*E_p_*–*E_r_*) for Q235 carbon steel in WO_4_^2−^ + Cl^−^, MoO_4_^2−^ + Cl^−^ and WO_4_^2−^ + MoO_4_^2−^ + Cl^−^ solutions.

Solution	*E_c_* (V_SCE_)	*i_c_* (µA/cm^2^)	*E_p_* (V_SCE_)	*E_r_* (V_SCE_)	*E_p_–E_r_* (V_SCE_)
WO_4_^2−^ + Cl^−^	−0.40	8.16	0.01	−0.25	0.35
MoO_4_^2−^ + Cl^−^	−0.39	7.52	0.09	−0.05	0.14
WO_4_^2−^ + MoO_4_^2−^ + Cl^−^	−0.38	6.83	0.17	0.06	0.09

**Table 3 materials-15-08986-t003:** Composition of the passive film on Q235 carbon steel surface formed in WO_4_^2−^ + Cl^−^, MoO_4_^2−^ + Cl^−^ and WO_4_^2−^ + MoO_4_^2−^ + Cl^−^ solutions.

Solution	Passive Film Composition	Description
WO_4_^2−^ + Cl^−^	FeOOH/Fe_2_O_3_, Fe(OH)_2_, Fe(OH)_3_, FeWO_4_, Fe_2_(WO_4_)_3_	FeWO_4_ plus Fe_2_(WO_4_)_3_ film
MoO_4_^2−^ + Cl^−^	FeOOH/Fe_2_O_3_, Fe(OH)_2_, Fe(OH)_3_, Fe_2_(MoO_4_)_3_	Fe_2_(MoO_4_)_3_ film
WO_4_^2−^ + MoO_4_^2−^ + Cl^−^	FeOOH/Fe_2_O_3_, Fe(OH)_2_, Fe(OH)_3_, Fe_2_(WO_4_)_3_, Fe_2_(MoO_4_)_3_	Fe_2_(WO_4_)_3_ plus Fe_2_(MoO_4_)_3_ film

**Table 4 materials-15-08986-t004:** Fitted values of *CPE_o_*, *R_o_*, *CPE_i_* and *R_i_* for Q235 carbon steel in WO_4_^2−^ + Cl^−^, MoO_4_^2−^ + Cl^−^ and WO_4_^2−^ + MoO_4_^2−^ + Cl^−^ solutions.

Solution	*CPE_o_* (μF/cm^2^)	*R_o_* (kΩ cm^2^)	*CPE_i_* (μF/cm^2^)	*R_i_* (kΩ cm^2^)
WO_4_^2−^ + Cl^−^	372.58	89.58	26.43	5.44
MoO_4_^2−^ + Cl^−^	258.47	148.73	25.07	5.13
WO_4_^2−^ + MoO_4_^2−^ + Cl^−^	140.21	179.58	23.92	5.67

**Table 5 materials-15-08986-t005:** Fitted values of *N_D_* and *U_f_* for Q235 carbon steel in WO_4_^2−^ + Cl^−^, MoO_4_^2−^ + Cl^−^ and WO_4_^2−^ + MoO_4_^2−^ + Cl^−^ solutions.

Solution	*N_D_* (cm^−3^)	*U_f_* (V_SCE_)
WO_4_^2−^ + Cl^−^	4.16 × 10^18^	−0.08
MoO_4_^2−^ + Cl^−^	3.83 × 10^18^	−0.05
WO_4_^2−^ + MoO_4_^2−^ + Cl^−^	1.92 × 10^18^	−0.01

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
