# Peer review of "The Synergistic Inhibitions of Tungstate and Molybdate Anions on Pitting Corrosion Initiation for Q235 Carbon Steel in Chloride Solution"

_materials, 2022, doi:10.3390/ma15248986_

Round 1
Reviewer 1 Report
The manuscript describes the study of the synergistic inhibitions effect of tungstate and molybdate on the pitting corrosion initiation of Q235 carbon steel in chloride solution. The applied methods are generally appropriate. Specific comments follow.
1. The introduction should be more theoretically supported. A review and main conclusions of the previous study of similar compounds and their action mechanisms, on the tested steel Q235, are missing.
2. What was the dimension of the surface area of the working electrode?
3. Figure 2: How exactly was the current density calculated using the Tafel slope? The passivated surface does not have the active dissolution region which is however required by the Tafel method.
4. Results of polarisation and EIS measurements in chloride solution without WO42- and MoO42- should be presented due to the comparison.
Author Response
Thank you very much for your comments, which are very professional and helpful.
The four comments of Reviewer 1 have been replied as follows, and the related contents of four comments have also been rewritten or supplemented in the revised manuscript of materials-2058256_R1. Please check them! If there is any question, please do not hesitate to let us know.

Reviewer 2 Report
Introduction:
The corrosion, especially the pitting corrosion of Q235 carbon steel in Cl- containing solution should be reviewed in the introduction.
The reason for selecting the Q235 steel should be justified.
Why these concentrations of Cl- ions and inhibitors were selected??
Materials and methods:
To clarify the inhibition role of the inhibitors, the corrosion behavior of the steel should be assessed in an inhibitor-free solution (solution containing only 0.1 mM NaCl).
Some standard methods can be used for assessing the pitting corrosion resistance such as cyclic polarization. If the current work aims to emphasize the pitting corrosion resistance it is recommended to use these methods.
Line 122: why a different Ee of Fe oxidation has been observed?
Line 201: why this behavior has been seen? This section needs more discussion.
Lines 225-227: more discussion should be added. Why the oxidation ability is important and its mechanism should be discussed in detail.
Table 4: other EEC model elements' values ( CPE and Cdl ) should be added and discussed.
Author Response
Thank you very much for your comments, which are very professional and helpful.
The nine comments of Reviewer 2 have been replied as follows, and the related contents of nine comments have also been rewritten or supplemented in the revised manuscript of materials-2058256_R1. Please check them! If there is any question, please do not hesitate to let us know.

Round 2
Reviewer 1 Report
The quality is improved as the authors have revised the
manuscript according to reviewers' comments, it can be accepted.